# On Fast Adversarial Robustness Adaptation in Model-Agnostic Meta-Learning

**Ren Wang**[1,4] *   **Kaidi Xu**[2]   **Sijia Liu**[3,5] †   **Pin-Yu Chen**[3]   **Tsui-Wei Weng**[3]   **Chuang Gan**[3]
**Meng Wang**[1]
[1]Rensselaer Polytechnic Institute, USA
[2]Northeastern University, USA
[3]MIT-IBM Watson AI Lab, IBM Research, USA
[4]University of Michigan, USA
[5]Michigan State University, USA

## Abstract

Model-agnostic meta-learning (MAML) has emerged as one of the most successful meta-learning techniques in few-shot learning. It enables us to learn a *meta-initialization* of model parameters (that we call *meta-model*) to rapidly adapt to new tasks using a small amount of labeled training data. Despite the generalization power of the meta-model, it remains elusive that how *adversarial robustness* can be maintained by MAML in few-shot learning. In addition to generalization, robustness is also desired for a meta-model to defend adversarial examples (attacks). Toward promoting adversarial robustness in MAML, we first study *when* a robustness-promoting regularization should be incorporated, given the fact that MAML adopts a bi-level (fine-tuning vs. meta-update) learning procedure. We show that robustifying the meta-update stage is sufficient to make robustness adapted to the task-specific fine-tuning stage even if the latter uses a standard training protocol. We also make additional justification on the acquired robustness adaptation by peering into the interpretability of neurons' activation maps. Furthermore, we investigate *how* robust regularization can *efficiently* be designed in MAML. We propose a general but easily-optimized robustness-regularized meta-learning framework, which allows the use of unlabeled data augmentation, fast adversarial attack generation, and computationally-light fine-tuning. In particular, we for the first time show that the auxiliary contrastive learning task can enhance the adversarial robustness of MAML. Finally, extensive experiments are conducted to demonstrate the effectiveness of our proposed methods in robust few-shot learning. Codes are available at https://github.com/wangren09/MetaAdv.

## 1 Introduction

Meta-learning, which can offer fast generalization adaptation to unseen tasks (Thrun & Pratt, 2012; Novak & Gowin, 1984), has widely been studied from model- and metric-based methods (Santoro et al., 2016; Munkhdalai & Yu, 2017; Koch et al., 2015; Snell et al., 2017) to optimization-based methods (Ravi & Larochelle, 2016; Finn et al., 2017; Nichol et al., 2018). In particular, model-agnostic meta-learning (MAML) (Finn et al., 2017) is one of the most intriguing bi-level optimization-based meta-learning methods designed for fast-adapted few-shot learning. That is, the learnt meta-model can rapidly be generalized to unforeseen tasks with only a small amount of data. It has successfully been applied to use cases such as object detection (Wang et al., 2020), medical image analysis (Maicas et al., 2018), and language modeling (Huang et al., 2018).

In addition to generalization-ability, recent works (Yin et al., 2018; Goldblum et al., 2019; Xu et al., 2020) investigated MAML from another fundamental perspective, *adversarial robustness*, given by the capabilities of a model defending against adversarially perturbed inputs (known as adversarial

---

*Corresponding: Ren Wang (wangren348117609@gmail.com, renwang@umich.edu).
†Work is done at the MIT-IBM Watson AI Lab

examples/attacks) (Goodfellow et al., 2014; Xu et al., 2019b). The challenge of lacking robustness of deep learning (DL) models has gained increasing interest and attention. And there exists a proactive arm race between adversarial attack and defense; see overview in (Carlini et al., 2019; Hao-Chen et al., 2020).

There have existed many defensive methods in the context of standard model training, e.g., (Madry et al., 2017; Zhang et al., 2019b; Wong et al., 2020; Carmon et al., 2019; Stanforth et al., 2019; Xu et al., 2019a), however, few work studied *robust MAML* except (Yin et al., 2018; Goldblum et al., 2019) to the best of our knowledge. And tackling such a problem is more challenging than robustifying the standard model training, since MAML contains a bi-leveled learning procedure in which the meta-update step (outer loop) optimizes a task-agnostic initialization of model parameters while the fine-tuning step (inner loop) learns a task-specific model instantiation updated from the common initialization. Thus, it remains elusive *when* (namely, at which learning stage) and *how* robust regularization should be promoted to strike a graceful balance between generalization/robustness and computation efficiency. Note that neither the standard MAML (Finn et al., 2017) nor the standard robust training (Madry et al., 2017; Zhang et al., 2019b) is as easy as normal training. Besides the algorithmic design in robust MAML, it is also important to draw in-depth explanation and analysis on *why* adversarial robustness can efficiently be gained in MAML. In this work, we aim to re-visit the problem of adversarial robustness in MAML (Yin et al., 2018; Goldblum et al., 2019) and make affirmative answers to the above questions on *when*, *how* and *why*.

**Contributions**    Compared to the existing works (Yin et al., 2018; Goldblum et al., 2019), we make the following contributions:

• Given the fact that MAML is formed as a bi-level learning procedure, we show and explain why regularizing adversarial robustness at the meta-update level is sufficient to offer fast and effective robustness adaptation on few-shot test tasks.

• Given the fact that either MAML or robust training alone is computationally intensive, we propose a general but efficient robustness-regularized meta-learning framework, which allows the use of unlabeled data augmentation, fast (one-step) adversarial example generation during meta-updating, and partial model training during fine-tuning (only fine-tuning the classifier's head).

• We for the first time show that the use of unlabeled data augmentation, particularly introducing an auxiliary contrastive learning task, can provide additional benefits on adversarial robustness of MAML in the low data regime, $2\%$ robust accuracy improvement and $9\%$ clean accuracy improvement over the state-of-the-art robust MAML method (named as *adversarial querying*) in (Goldblum et al., 2019).

**Related work**    To train a standard model (instead of a meta-model), the most effective robust training methods include adversarial training (Madry et al., 2017), TRADES that places a theoretically-grounded trade-off between accuracy and robustness (Zhang et al., 2019b), and their many variants such as fast adversarial training methods (Shafahi et al., 2019; Zhang et al., 2019a; Wong et al., 2020; Andriushchenko & Flammarion, 2020), semi-supervised robust training (Carmon et al., 2019; Stanforth et al., 2019), adversarial transfer learning and certifiably robust training (Wong & Kolter, 2017; Dvijotham et al., 2018). Moreover, recent works (Hendrycks et al., 2019; Chen et al., 2020a; Shafahi et al., 2020; Chan et al., 2020; Utrera et al., 2020; Salman et al., 2020) studied the transferability of robustness in in the context of transfer learning and representation learning. However, the aforementioned standard robust training methods are not directly applicable to MAML in few-shot learning considering MAML's bi-leveled optimization nature.

A few recent works studied the problem of adversarial training in the context of MAML (Goldblum et al., 2019; Yin et al., 2018). Yin et al. (2018) considered the robust training in both fine-tuning and meta-update steps, which is unavoidably computationally expensive and difficult in optimization. The most relevant work to ours is (Goldblum et al., 2019), which proposed adversarial querying (AQ) by integrating adversarial training with MAML. Similar to ours, AQ attempted to robustify meta-update only to gain sufficient robustness. However, it lacks explanation for the rationale behind that. We will show that AQ can also be regarded as a special case of our proposed robustness-promoting MAML framework. Most important, we make a more in-depth study with novelties summarized in **Contributions**.

Another line of research relevant to ours is efficient MAML, e.g., (Raghu et al., 2019; Song et al., 2019; Su et al., 2019), where the goal is to improve the computation efficiency and/or the generalization of MAML. In (Song et al., 2019), gradient-free optimization was leveraged to alleviate the need of second-order derivative information during meta-update. In (Raghu et al., 2019), MAML was simplified by removing the fine-tuning step over the representation block of a meta-model. It was shown that such a simplification is surprisingly effective without losing generalization-ability. In (Su et al., 2019), a self-supervised representation learning task was augmented to the meta-updating objective and resulted in a meta-model with improved generalization. Although useful insights were gained from MAML in the aforementioned works, none of them took adversarial robustness into account.

## 2 PRELIMINARIES AND PROBLEM STATEMENT

In this section, we first review model-agnostic meta learning (MAML) (Finn et al., 2017) and adversarial training (Madry et al., 2017), respectively. We then motivate the setup of robustness-promoting MAML and demonstrate its challenges in design when integrating MAML with robust regularization.

**MAML**    MAML attempts to learn an initialization of model parameters (namely, a meta-model) so that a new few-shot task can quickly and easily be tackled by fine-tuning this meta-model over a small amount of labeled data. The characteristic signature of MAML is its *bi-level* learning procedure, where the fine-tuning stage forms a task-specific *inner loop* while the meta-model is updated at the *outer loop* by minimizing the validation error of fine-tuned models over cumulative tasks. Formally, consider $N$ few-shot learning tasks $\{\mathcal{T}_i\}_{i=1}^N$, each of which has a fine-tuning data set $\mathcal{D}_i$ and a validation set $\mathcal{D}_i'$, where $\mathcal{D}_i$ is used in the *fine-tuning* stage and $\mathcal{D}_i'$ is used in the *meta-update* stage. Here the superscript $(\prime)$ is preserved to indicate operations/parameters at the meta-upate stage. MAML is then formulated as the following bi-level optimization problem (Finn et al., 2017):

$$
\begin{aligned}
\underset{\mathbf{w}}{\text{minimize}} \quad & \tfrac{1}{N} \sum_{i=1}^N \ell_i'(\mathbf{w}_i'; \mathcal{D}_i') \\
\text{subject to} \quad & \mathbf{w}_i' = \arg\min_{\mathbf{w}_i} \ell_i(\mathbf{w}_i; \mathcal{D}_i, \mathbf{w}), \ \forall i \in [N]
\end{aligned}
\tag{1}
$$

where $\mathbf{w}$ denotes the meta-model to be designed, $\mathbf{w}_i'$ is the $\mathcal{T}_i$-specific fine-tuned model, $\ell_i'(\mathbf{w}_i'; \mathcal{D}_i')$ represents the validation error using the fine-tuned model, $\ell_i(\mathbf{w}_i; \mathcal{D}_i, \mathbf{w})$ denotes the training error when fine-tuning the task-specific model parameters $\mathbf{w}_i$ using the task-agnostic initialization $\mathbf{w}$, and for ease of notation, $[K]$ represents the integer set $\{1, 2, \ldots, K\}$. In (1), the objective function and the constraint correspond to the meta-update stage and fine-tuning stage, respectively. The bi-level optimization problem is challenging because each constraint calls an inner optimization oracle, which is typically instantiated into a $K$-step gradient descent (GD) based solver:

$$
\mathbf{w}_i^{(k)} = \mathbf{w}_i^{(k-1)} - \alpha \nabla_{\mathbf{w}_i} \ell_i(\mathbf{w}_i^{(k-1)}; \mathcal{D}_i, \mathbf{w}), \ k \in [K], \ \text{with } \mathbf{w}_i^{(0)} = \mathbf{w}.
$$

We note that even with the above simplified fine-tuning step, updating the meta-model $\mathbf{w}$ still requires the second-order derivatives of the objective function of (1) with respect to (w.r.t.) $\mathbf{w}$.

**Adversarial training**    The min-max optimization based adversarial training (AT) is known as one of the most powerful defense methods to obtain a robust model against adversarial attacks (Madry et al., 2017). We summarize AT and its variants through the following robustness-regularized optimization problem:

$$
\underset{\mathbf{w}}{\text{minimize}} \quad \lambda \mathbb{E}_{(\mathbf{x},y) \in \mathcal{D}} \left[ \ell(\mathbf{w}; \mathbf{x}, y) \right] + \underbrace{\mathbb{E}_{(\mathbf{x},y) \in \mathcal{D}} [\underset{\|\boldsymbol{\delta}\|_\infty \leq \epsilon}{\text{maximize}} \, g(\mathbf{w}; \mathbf{x} + \boldsymbol{\delta}, y)]}_{\mathcal{R}(\mathbf{w}; \mathcal{D})},
\tag{2}
$$

where $\ell(\mathbf{w}; \mathbf{x}, y)$ denotes the prediction loss evaluated at the point $\mathbf{x}$ with label $y$, $\lambda \geq 0$ is a regularization parameter, $\boldsymbol{\delta}$ denotes the input perturbation variable within the $\ell_\infty$-norm ball of radius $\epsilon$, $g$ represents the robust loss evaluated at the model $\mathbf{w}$ at the perturbed example $\mathbf{x} + \boldsymbol{\delta}$ given the true label $y$, and for ease of notation, let $\mathcal{R}(\mathbf{w}; \mathcal{D})$ denote the robust regularization function for model $\mathbf{w}$ under the data set $\mathcal{D}$. In the rest of the paper, we consider two specifications of $\mathcal{R}$: (a) *AT regularization* (Madry et al., 2017), where we set $g = \ell$ and $\lambda = 0$; (b) *TRADES regularization* (Zhang et al., 2019b), where we define $g$ as the cross-entropy between the distribution of prediction probabilities at the perturbed example $(\mathbf{x} + \boldsymbol{\delta})$ and that at the original sample $\mathbf{x}$.

**Robustness-promoting MAML**  Integrating MAML with AT is a natural solution to enhance adversarial robustness of a meta-model in few-shot learning. However, this seemingly simple scheme is in fact far from trivial, and there exist three critical roadblocks as elaborated below.

*First*, it remains elusive at which stage (fine-tuning or meta-update) robustness can most effectively be gained for MAML. Based on (1) and (2), we can cast this problem as a unified optimization problem that augments the MAML loss with the robust regularization under two degrees of freedom characterized by two hyper-parameters $\gamma_{\text{out}} \geq 0$ and $\gamma_{\text{in}} \geq 0$:

$$\begin{aligned} \underset{\mathbf{w}}{\text{minimize}} \quad & \frac{1}{N}\sum_{i=1}^{N}[\ell'_i(\mathbf{w}'_i;\mathcal{D}'_i) + \gamma_{\text{out}}\mathcal{R}_i(\mathbf{w}'_i;\mathcal{D}'_i)] \\ \text{subject to} \quad & \mathbf{w}'_i = \arg\min_{\mathbf{w}_i}[\ell_i(\mathbf{w}_i;\mathcal{D}_i,\mathbf{w}) + \gamma_{\text{in}}\mathcal{R}_i(\mathbf{w}_i;\mathcal{D}_i)], \ \forall i \in [N]. \end{aligned} \tag{3}$$

Here $\mathcal{R}_i$ denotes the task-specific robustness regularizer, and the choice of $(\gamma_{\text{in}}, \gamma_{\text{out}})$ determines the specific scenario of robustness-promoting MAML. Clearly, the direct application is to set $\gamma_{\text{in}} > 0$ and $\gamma_{\text{out}} > 0$, that is, both fine-tuning and meta-update steps would be carried out using robust training, which calls additional loops to generate adversarial examples. Thus, this would make computation most intensive. Spurred by that, we ask: *Is it possible to achieve a robust meta-model by incorporating robust regularization into only either meta-update or fine-tuning step (corresponding to $\gamma_{\text{in}} = 0$ or $\gamma_{\text{out}} = 0$)?*

*Second*, both MAML in (1) and AT in (2) are challenging bi-level optimization problems which need to call inner optimization routines for fine-tuning and attack generation, respectively. Thus, we ask whether or not the computationally-light alternatives of inner solvers, e.g., partial fine-tuning (Raghu et al., 2019) and fast attack generation (Wong et al., 2020), can promise adversarial robustness in few-shot learning.

*Third*, it has been shown that adversarial robustness can benefit from semi-supervised learning by leveraging (unlabeled) data augmentation (Carmon et al., 2019; Stanforth et al., 2019). Spurred by that, we further ask: *Is it possible to generalize robustness-promoting MAML to the setup of semi-supervised learning for improved accuracy-robustness tradeoff?*

## 3 WHEN TO INCORPORATE ROBUST REGULARIZATION IN MAML?

In this section, we evaluate at which stage adversarial robustness can be gained during meta-training. We will provide insights and step-by-step investigations to show when to incorporate robust training in MAML and why it works. Based on (3), we focus on two robustness-promoting meta-training protocols. **(a)** R-MAML$_{\text{both}}$, where robustness regularization applied to *both* fine-tuning and meta-update steps with $\gamma_{\text{in}}, \gamma_{\text{out}} > 0$; **(b)** R-MAML$_{\text{out}}$, where robust regularization applied to *meta-update only*, i.e., $\gamma_{\text{in}} = 0$ and $\gamma_{\text{out}} > 0$. Compared to R-MAML$_{\text{both}}$, R-MAML$_{\text{out}}$ is more user-friendly since it allows the use of *standard* fine-tuning over the learnt robust meta-model when tackling unseen few-shot test tasks (known as meta-testing). In what follows, we will show that even if R-MAML$_{\text{out}}$ does not use robust regularization in fine-tuning, it is sufficient to warrant the transferability of meta-model's robustness to downstream fine-tuning tasks.

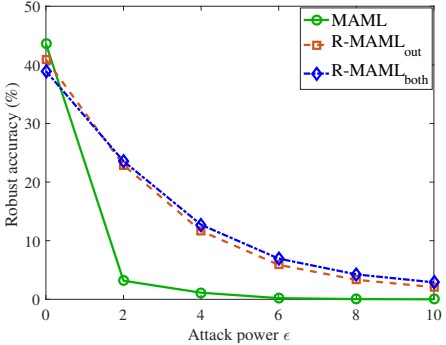

Figure 1: RA of meta-models trained by standard MAML, R-MAML$_{\text{both}}$ and R-MAML$_{\text{out}}$ versus PGD attacks of different perturbation sizes during meta-testing. Results show that robustness regularized meta-update with standard fine-tuning (namely, R-MAML$_{\text{out}}$) has already been effective in promotion of robustness.

**All you need is robust meta-update during meta-training**  To study this claim, we solve problem (3) using R-MAML$_{\text{both}}$ and R-MAML$_{\text{out}}$ respectively in the 5-way 1-shot learning setup, where 1 data sample at each of 5 randomly selected MiniImagenet classes (Ravi & Larochelle, 2016) constructs a learning task. Throughout this section, we specify $\mathcal{R}_i$ in (3) as the AT regularization, which calls a 10-step projected gradient descent (PGD) attack generation method with $\epsilon = 2/255$ in its inner maximization subroutine given by (2). We refer readers to Section 6 for more implementation details.

We find that the meta-model acquired by R-MAML$_{\text{out}}$ yields *nearly the same robust accuracy* (RA) as R-MAML$_{\text{both}}$ against various PGD attacks generated at the testing phase using different perturbation sizes $\epsilon = \{0, 2, \ldots, 10\}/255$ as shown in Figure 1. Unless specified otherwise, we evaluate the performance of the meta-learning schemes over 2400 random unseen 5-way 1-shot test tasks. We also note that RA under $\epsilon = 0$ becomes the standard accuracy (SA) evaluated using benign (unperturbed) test examples. It is clear from Figure 1 that both R-MAML$_{\text{out}}$ and R-MAML$_{\text{both}}$ can yield significantly better RA than MAML with slightly worse SA. It is also expected that RA decreases as the attack power $\epsilon$ increases.

Spurred by experiment results in Figure 1, we hypothesize that the promotion of robustness in meta-update *alone* (i.e. R-MAML$_{\text{out}}$) is already sufficient to offer *robust representation*, over which fine-tuned models can preserve robustness to downstream tasks. In what follows, we justify the above hypothesis from two perspectives: (i) explanation of learned neuron's representation and (ii) resilience of learnt robust meta-model to different fine-tuning schemes at the meta-testing phase.

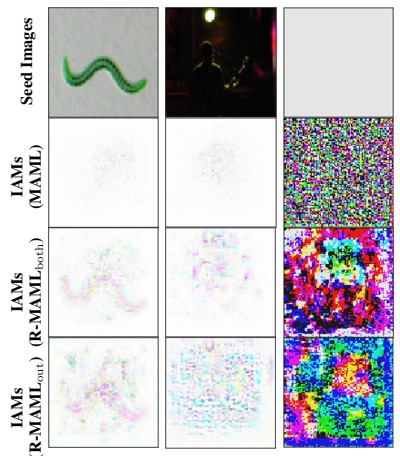

Figure 2: Visualization of a randomly selected neuron's inverted input attribution maps (IAMs) under different meta-models. The first row shows the seed images. The second-fourth rows show IAMs corresponding to models trained by MAML, R-MAML$_{\text{both}}$, and R-MAML$_{\text{out}}$, respectively. Except MAML, R-MAML$_{\text{both}}$ and R-MAML$_{\text{out}}$ all catch high-level features from the data.

**(i) Learned signature of neuron's representation** It is recently shown in (Engstrom et al., 2019) that a robust model exhibits perceptually-aligned neuron activation maps, which are not present if the model lacks adversarial robustness. To uncover such a signature of robustness, a feature inversion technique (Engstrom et al., 2019) is applied to finding an inverted *input attribution map* (IAM) that maximizes neuron's activation. Based on that, we examine if R-MAML$_{\text{both}}$ and R-MAML$_{\text{out}}$ can similarly generate explainable inverted images from the learned neuron's representation. We refer readers to Appendix 2 for more details on feature inversion from neuron's activation.

In our experiment, we indeed find that both R-MAML$_{\text{both}}$ and R-MAML$_{\text{out}}$ yield similar IAMs inverted from neuron's activation at different input examples, as plotted in Figure 2. More intriguingly, the learnt IAMs characterize the contour of objects existed in input images, and accompanied by the learnt high-level features, e.g., colors. In contrast, the IAMs of MAML lack such an interpretability. The observations from the interpretability of neurons' representation justify why R-MAML$_{\text{out}}$ is as effective as R-MAML$_{\text{both}}$ and why MAML does not preserve robustness.

**(ii) Robust meta-update provides robustness adaptation without additional adversarial fine-tuning at meta-testing** Meta-testing includes only the fine-tuning stage. Therefore, we need to explore if standard fine-tuning is enough to maintain the robustness. Suppose that R-MAML$_{\text{out}}$ is adopted as the meta-training method to solve problem (3), we then ask if robustness-regularized meta-testing strategy can improve the robustness of fine-tuned model at downstream tasks. Surprisingly, we find

Table 1: Comparison of different strategies in meta-testing on R-MAML$_{\text{out}}$: (a) standard fine-tuning (S-FT), (b) adversarial fine-tuning (A-FT).

|  | S-FT | A-FT |
|---|---|---|
| SA | 40.9% | 39.6% |
| RA | 22.9% | 23.5% |

that making an additional effort to adversarially fine-tune the meta-model (trained by R-MAML$_{\text{out}}$) during testing does *not* provide an obvious robustness improvement over the standard fine-tuning scheme during testing (Table 1). This consistently implies that robust meta-update (R-MAML$_{\text{out}}$) is sufficient to render intrinsic robustness in its learnt meta-model regardless of fine-tuning strategies used at meta-testing. Figure S1 in Appendix 3 provides evidence that the visualization difference is small between before standard fine-tuning and after standard fine-tuning.

**Adversarial querying (AQ) (Goldblum et al., 2019): A special case of R-MAML$_{\text{out}}$** The recent work (Goldblum et al., 2019) developed AQ to improve adversarial robustness in few-shot learning.

AQ can be regarded as a special case of R-MAML$_{\text{out}}$ with $\gamma_{\text{in}} = 0$ but setting $\gamma_{\text{out}} = \infty$ in (3). That is, the meta-update is overridden by the AT regularization. We find that AQ yields about 2% RA improvement over R-MAML$_{\text{out}}$, which uses $\gamma_{\text{out}} = 0.2$ in (3). However, AQ leads to 11% degradation in SA, and thus makes a much poorer robustness-accuracy tradeoff than our proposed R-MAML$_{\text{out}}$. We refer readers to Table 2 for comparison of the proposed R-MAML$_{\text{out}}$ with other training baselines. Most importantly, different from (Goldblum et al., 2019), we provide insights on why R-MAML$_{\text{out}}$ is effective in promoting adversarial robustness from meta-update to fine-tuning.

## 4 COMPUTATIONALLY-EFFICIENT ROBUSTNESS-REGULARIZED MAML

In this section, we study if the proposed R-MAML$_{\text{out}}$ can further be improved to ease of optimization given the two computation difficulties in (3): (a) bi-leveled meta-learning, and (b) the need of inner maximization to find the worst-case robust regularization. To tackle either problem alone, there have been efficient solution methods proposed recently. In (Raghu et al., 2019), an almost-no-inner-loop (ANIL) fine-tuning strategy was proposed, where fine-tuning is only applied to the task-specific classification head following a frozen representation network inherited from the meta-model. Moreover, in (Wong et al., 2020), a fast gradient sign method (FGSM) based attack generator was leveraged to improve the efficiency of AT without losing its adversarial robustness. Motivated by (Raghu et al., 2019; Wong et al., 2020), we ask if integrating R-MAML$_{\text{out}}$ with ANIL and/or FGSM can improve the training efficiency but preserves the robustness and generalization-ability of a meta-model learnt from R-MAML$_{\text{out}}$.

**R-MAML$_{\text{out}}$ meets ANIL and FGSM** We decompose the meta-model $\mathbf{w} = [\mathbf{w}_{\text{r}}, \mathbf{w}_{\text{c}}]$ into two parts: representation encoding network $\mathbf{w}_{\text{r}}$ and classification head $\mathbf{w}_{\text{c}}$. In R-MAML$_{\text{out}}$, namely, (3) with $\gamma_{\text{in}} = 0$, ANIL suggests to only fine-tune $\mathbf{w}_{\text{c}}$ over a specific task $\mathcal{T}_i$. This leads to

$$\mathbf{w}'_{\text{c},i} = \arg\min_{\mathbf{w}_{\text{c},i}} \ell_i(\mathbf{w}_{\text{c},i}, \mathbf{w}_{\text{r}}; \mathcal{D}_i, \mathbf{w}), \text{ with } \mathbf{w}'_{\text{r},i} = \mathbf{w}_{\text{r}}. \quad \text{(ANIL)}$$

In ANIL, the initialized representation network $\mathbf{w}_{\text{r}}$ keeps intact during task-specific fine-tuning, which thus saves the computation cost. Furthermore, if FGSM is used in R-MAML$_{\text{out}}$, then the robustness regularizer $\mathcal{R}$ defined in (2) reduces to

$$\mathcal{R}(\mathbf{w}; \mathcal{D}) = \mathbb{E}_{(\mathbf{x},y) \in \mathcal{D}}[g(\mathbf{w}; \mathbf{x} + \boldsymbol{\delta}^*(\mathbf{x}), y)], \quad \boldsymbol{\delta}^*(\mathbf{x}) = \boldsymbol{\delta}_0 + \epsilon \nabla_{\mathbf{x}} g(\mathbf{w}; \mathbf{x}, y), \quad \text{(FGSM)}$$

where $\boldsymbol{\delta}_0$ is an initial point randomly drawn from a uniform distribution over the interval $[-\epsilon, \epsilon]$. Note that in the original implementation of robust regularization $\mathcal{R}$, a multi-step projected gradient ascent (PGA) is typically used to optimize the sample-wise adversarial perturbation $\boldsymbol{\delta}(\mathbf{w})$. By contrast, FGSM only uses one-step PGA in attack generation and thus improves the computation efficiency.

In Table 2, we study two computationally-light alternatives of R-MAML$_{\text{out}}$, R-MAML$_{\text{out}}$ with ANIL (R-MAML$_{\text{out}}$-ANIL) and R-MAML$_{\text{out}}$ with FGSM (R-MAML$_{\text{out}}$-FGSM). Compared to R-MAML$_{\text{out}}$, we find that although R-MAML$_{\text{out}}$-FGSM takes less computation time, it yields even better RA with slightly worse SA. By contrast, R-MAML$_{\text{out}}$-ANIL yields the least computation cost but the worst SA

Table 2: Performance of computation-efficient alternatives of R-MAML$_{\text{out}}$ in SA, RA and computation time per epoch (in minutes).

| | SA | RA | Time |
|---|---|---|---|
| MAML | 43.6% | 3.17% | 42min |
| AQ (Goldblum et al., 2019) | 29.6% | 24.9% | 52min |
| R-MAML$_{\text{out}}$ | 40.9% | 22.9% | 54min |
| R-MAML$_{\text{out}}$-ANIL | 37.46% | 22.7% | 36min |
| R-MAML$_{\text{out}}$-FGSM | 40.82% | 23.04% | 44min |

and RA. For comparison, we also present the performance of the adversarial meta-learning baseline AQ (Goldblum et al., 2019). As we can see, AQ promotes the adversarial robustness at the cost of a significant SA drop, e.g., 7.56% worse than R-MAML$_{\text{out}}$-ANIL. Overall, the application of FGSM to R-MAML$_{\text{out}}$ provides the most graceful tradeoff between the computation cost and the standard and robust accuracies. In the rest of the paper, unless specified otherwise we will use FGSM in R-MAML$_{\text{out}}$.

## 5 SEMI-SUPERVISED ROBUSTNESS-PROMOTING MAML

Given our previous solutions to *when* (Sec. 3) and *how* (Sec. 4) a robust regularization could effectively be promoted in few-shot learning, we next ask: Is it possible to further improve our proposal R-MAML$_\text{out}$ by leveraging *unlabeled data*? Such a question is motivated from two aspects. First, the use of unlabeled data augmentation could be a key momentum to improve the robustness-accuracy tradeoff (Carmon et al., 2019; Stanforth et al., 2019). Second, the recent success in self-supervised contrastive representation learning (Chen et al., 2020b; He et al., 2020) demonstrates the power of multi-view (unlabeled) data augmentation to acquire discriminative and generalizable visual representations, which can guide down-stream supervised learning. In what follows, we propose an extension of R-MAML$_\text{out}$ applicable to semi-supervised learning with unlabeled data augmentation.

**R-MAML$_\text{out}$ with TRADES regularization.** We recall from (2) that the robust regularization $\mathcal{R}$ can also be specified by TRADES (Zhang et al., 2019b), which relies only on the prediction logits of benign and adversarial examples (rather than the training label), and thus lends itself to the application of unlabeled data. Spurred by that, we propose R-MAML$_\text{out}$-TRADES, which is a variant of R-MAML$_\text{out}$ using the unlabeled data augmented TRADES regularization. To perform data augmentation in experiments, we follow (Carmon et al., 2019) to mine additional (unlabeled) data with the same amount of MiniImagenet data from the original ImageNet data set. For clarity, we call R-MAML$_\text{out}$ using TRADES or AT regularization (but without unlabeled data augmentation) R-MAML$_\text{out}$(TRADES) or R-MAML$_\text{out}$(AT).

We find that with the help of unlabeled data, R-MAML$_\text{out}$-TRADES improves the accuracy-robustness tradeoff over its supervised counterpart R-MAML$_\text{out}$ using either AT or TRADES regularization (Figure 3). Compared to R-MAML$_\text{out}$, R-MAML$_\text{out}$-TRADES yields consistently better RA against different attack strength $\epsilon \in \{2, \dots, 10\}/255$ during testing. Interestingly, the improvement becomes more significant as $\epsilon$ increases. As $\epsilon = 0$, RA is equivalent to SA, and we observe that the superior performance of R-MAML$_\text{out}$-TRADES in RA bears a slight degradation in SA compared to R-MAML$_\text{out}$(TRADES) and R-MAML$_\text{out}$(AT), which indicates the robustness-accuracy tradeoff. Figure S2 in Appendix 5 provides an additional evidence that R-MAML$_\text{out}$-TRADES has the ability to defend stronger attacks than R-MAML$_\text{out}$, and proper unlabeled data augmentation can further improve the accuracy-robustness tradeoff in MAML.

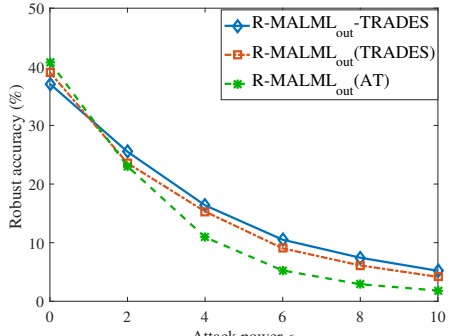

Figure 3: RA versus (testing-phase) PGD attacks at different values of perturbation strength $\epsilon$. Here the robust models are trained by different variants of R-MAML$_\text{out}$, including R-MAML$_\text{out}$-TRADES (with unlabeled data augmentation), R-MAML$_\text{out}$ using AT regularization but no data augmentation (R-MAML$_\text{out}$(AT)), and R-MAML$_\text{out}$ using TRADES regularization but no data augmentation (R-MAML$_\text{out}$(TRADES)).

**R-MAML$_\text{out}$ with contrastive learning (CL).** To improve adversarial robustness, many works, e.g., (Pang et al., 2019; Sankaranarayanan et al., 2017), also suggest that it is important to encourage robust semantic features that locally cluster according to class, namely, ensuring that features of samples in the same class will lie close to each other and away from those of different classes. The above suggestion aligns with the goals of contrastive learning (CL) (Chen et al., 2020b; Wang & Isola, 2020), which promotes (a) alignment (closeness) of features from positive data pairs, and (b) uniformity of feature distribution. Thus, we develop R-MAML$_\text{out}$-CL by integrating R-MAML$_\text{out}$ with CL.

Prior to defining R-MAML$_\text{out}$-CL, we first introduce CL and refer readers to (Chen et al., 2020b) for details. Given a data sample $\mathbf{x}$, CL utilizes its positive counterpart $\mathbf{x}^+$ given by a certain data transformation $t$, e.g., cropping and resizing, cut-out, and rotation, $\mathbf{x}^+ = t(\mathbf{x})$. The data pair $(\mathbf{x}, t(\mathbf{x}'))$ is then positive if $\mathbf{x} = \mathbf{x}'$, and negative otherwise. The *contrastive loss* is defined by

$$\ell_{\text{CL}}(\mathbf{w}_c; p^+) = \mathbb{E}_{(\mathbf{x}, \mathbf{x}^+) \sim p^+} \left[ -\log \frac{e^{\mathbf{r}(\mathbf{x}; \mathbf{w}_c)^T \mathbf{r}(\mathbf{x}^+; \mathbf{w}_c)/\tau}}{e^{\mathbf{r}(\mathbf{x}; \mathbf{w}_c)^T \mathbf{r}(\mathbf{x}^+; \mathbf{w}_c)/\tau} + \sum_{\mathbf{x}^- \sim p, (\mathbf{x}, \mathbf{x}^-) \notin p^+} \left[ e^{\mathbf{r}(\mathbf{x}; \mathbf{w}_c)^T \mathbf{r}(\mathbf{x}^-; \mathbf{w}_c)/\tau} \right]} \right],$$

where $\mathbf{x} \sim p$ denotes the data distribution, $p^+(\cdot,\cdot)$ is the distribution of positive pairs, $\mathbf{r}(\mathbf{x};\mathbf{w}_c)$ is the encoded representation of $\mathbf{x}$ extracted from the representation network $\mathbf{w}_c$, and $\tau > 0$ is a temperature parameter. The contrastive loss minimizes the distance of a positive pair among many negative pairs, namely, learns network representation with instance-wise discriminative power.

According to CL, we then augment the data used to train R-MAML$_{\text{out}}$ with their transformed counterparts. In addition, the *adversarial examples* generated during robust regularization can also be used as *additional views* of the original data, which in turn advance CL. Formally, we modify R-MAML$_{\text{out}}$, given by (3) with $\gamma_{\text{in}} = 0$, as

$$\begin{aligned} \underset{\mathbf{w}}{\text{minimize}} \quad & \tfrac{1}{N}\sum_{i=1}^{N}\left[\ell_i'(\mathbf{w}_i';\mathcal{D}_i') + \gamma_{\text{out}}\mathcal{R}_i(\mathbf{w}_i';\mathcal{D}_i') + \gamma_{\text{CL}}\ell_{\text{CL}}(\mathbf{w}_{c,i}';p_i^+ \cup p_i^{\text{adv}})\right] \\ \text{subject to} \quad & \mathbf{w}_i' = \arg\min_{\mathbf{w}_i}\ell_i(\mathbf{w}_i;\mathcal{D}_i,\mathbf{w}), \ \forall i \in [N], \end{aligned} \qquad (4)$$

where $\gamma_{\text{CL}} > 0$ is a regularization parameter associated with the contrastive loss, $p_i^+ \cup p_i^{\text{adv}}$ represents the distribution of positive data pairs constructed by the standard and adversarial views of $\mathcal{D}'$, and $\mathbf{w}_{c,i}'$ denotes the representation block of the model $\mathbf{w}_i'$.

In Table 3, we compare the SA/RA performance of R-MAML$_{\text{out}}$-CL with that of previously-suggested 3 variants of R-MAML$_{\text{out}}$ including the versions R-MAML$_{\text{out}}$(AT) and R-MAML$_{\text{out}}$(TRADES) without using unlabeled data, and the version with unlabeled data R-MAML$_{\text{out}}$-TRADES, as well as 2 baseline methods including standard MAML and adversarial querying (AQ) in few-shot learning (Goldblum et al., 2019). Note that we specify $\mathcal{R}_i$ in

Table 3: SA/RA performance of R-MAML$_{\text{out}}$-CL versus other variants of proposed R-MAML$_{\text{out}}$ and baselines.

|  | SA | RA |
|---|---|---|
| MAML | **43.6%** | 3.17% |
| AQ (Goldblum et al., 2019) | 29.6% | 24.9% |
| R-MAML$_{\text{out}}$(AT) (ours) | 40.82% | 23.04% |
| R-MAML$_{\text{out}}$(TRADES) (ours) | 39.06% | 23.56% |
| R-MAML$_{\text{out}}$-TRADES (ours) | 37.1% | 25.51% |
| R-MAML$_{\text{out}}$-CL (ours) | 38.60% | **26.81%** |

(4) as TRADES regularization for R-MAML$_{\text{out}}$-CL. We find that R-MAML$_{\text{out}}$-CL yields the best RA among all meta-learning methods, and improves SA over R-MAML$_{\text{out}}$-TRADES. In particular, the comparison with AQ shows that R-MAML$_{\text{out}}$-CL leads to 9% improvement in SA and 1.9% improvement in RA.

## 6 ADDITIONAL EXPERIMENTS

**Key facts of our implementation.** In the previous analysis, we consider 1-shot 5-way image classification tasks over MiniImageNet (Vinyals et al., 2016). And we use a four-layer convolutional neural network for few-shot learning (FSL). By default, we set the training attack strength $\epsilon = 2$, $\gamma_{\text{CL}} = 0.1$, and set

Table 4: Summary of baseline performance in SA and TA.

|  | SA | RA |
|---|---|---|
| MAML (FSL) | **43.6%** | 3.17% |
| AQ (FSL) (Goldblum et al., 2019) | 29.6% | **24.9%** |
| Supervised standard training (non-FSL) | 29.74% | 3.51% |
| Supervised AT (non-FSL) | 28.22% | 19.02% |

$\gamma_{\text{out}} = 5$ (TRADES), $\gamma_{\text{out}} = 0.2$ (AT) via a grid search. During meta-testing, a 10-step PGD attack with attack strength $\epsilon = 2$ is used to evaluate RA of the learnt meta-model over 2400 few-shot test tasks. We provide experiment details in Appendix 4.

**Summary of baselines.** We remark that in addition to MAML and AQ baselines, we also consider the other two baseline methods, supervised standard training over the entire dataset (non-FSL setting), and supervised AT over the entire dataset (non-FSL setting); see a summary in Table 4. The additional baselines demonstrate that robust adaptation in FSL is non-trivial as neither the supervised full AT or the full standard training can achieve satisfactory SA and RA.

**Experiments on Additional model architecture, datasets and FSL setups.** In Table S1 of Appendix 5, we provide additional experiments using ResNet18. In particular, R-MAML$_{\text{out}}$-CL leads to 13.94% SA improvement and 1.42% RA improvement over AQ. We also test our methods on CIFAR-FS (Bertinetto et al., 2018) and Omniglot (Lake et al., 2015), and provide the results in Table 5 and Figure S3, respectively (more details can be viewed in Appendix 6 and Appendix 7). The

Table 5: SA/RA performance of our proposed methods on CIFAR-FS (Bertinetto et al., 2018).

|  | 1-Shot 5-Way | | 5-Shot 5-Way | |
| --- | --- | --- | --- | --- |
|  | SA | RA | SA | RA |
| MAML | **51.07%** | 0.235% | **67.2%** | 0.225% |
| AQ (Goldblum et al., 2019) | 31.25% | 26.34% | 52.32% | 33.96% |
| R-MAML$_{out}$(AT) (ours) | 39.76% | 26.15% | 57.18% | 32.62% |
| R-MAML$_{out}$(TRADES) (ours) | 40.23% | 27.45% | 57.46% | 34.72% |
| R-MAML$_{out}$-TRADES (ours) | 40.59% | 28.06% | 57.62% | 34.76% |
| R-MAML$_{out}$-CL (ours) | 41.25% | **29.33%** | 57.95% | **35.30%** |

results show that our methods perform well on various datasets and outperform the baseline methods. On CIFAR-FS, we study 1-Shot 5-Way and 5-Shot 5-Way settings. As shown in Table 5, the use of unlabeled data augmentation (R-MAML$_{out}$-CL) on CIFAR-FS can provide $10\%$ (or $5.6\%$) SA improvement and $3\%$ (or $1.3\%$) RA improvement over AQ under the 1-Shot 5-Way (or 5-Shot 5-Way) setting. Furthermore, we conduct experiments in other FSL setups. On Omniglot, we compare R-MAML$_{out}$(TRADES) to AQ (Goldblum et al., 2019) in the 1-shot (5, 10, 15, 20)-Way settings. Figure S3 shows that R-MAML$_{out}$(TRADES) can always obtain better performance than AQ when the number of classes in each task varies.

## 7 CONCLUSION

In this paper, we study the problem of adversarial robustness in MAML. Beyond directly integrating MAML with robust training, we show and explain when a robust regularization should be promoted in MAML. We find that robustifying the meta-update stage via fast attack generation method is sufficient to achieve fast robustness adaptation without losing generalization and computation efficiency in general. To further improve our proposal, we for the first time study how unlabeled data help robust MAML. In particular, we propose using contrastive representation learning to acquire improved generalization and robustness simultaneously. Extensive experiments are provided to demonstrate the effectiveness of our approach and justify our insights on the adversarial robustness of MAML. In the future, we plan to establish the convergence rate analysis of robustness-aware MAML by leveraging bi-level and min-max optimization theories.

## ACKNOWLEDGEMENT

This work was supported by the Rensselaer-IBM AI Research Collaboration (`http://airc.rpi.edu`), part of the IBM AI Horizons Network (`http://ibm.biz/AIHorizons`).

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

SUPPLEMENTARY MATERIAL

# 1 FRAMEWORK OF R-MAML$_{\text{out}}$

Algorithm S1 shows the framework of R-MAML$_{\text{out}}$. The initial inputs include model weights $\mathbf{w}$, distribution of the training tasks $p(\mathcal{T})$, and the step sizes $\alpha, \beta_1, \beta_2$, which correspond to fine-tuning, clean meta-update, adversarial meta-update. Each batch contains multiple tasks that are sampled from the $p(\mathcal{T})$. $K$ is the number of gradient updates in fine-tuning. The adapted parameter $\mathbf{w}_i^{(K)}$ is used to generate adversarial validation data $\hat{\mathcal{D}}_i'$ from the clean validation data $\mathcal{D}_i'$ and to compute the loss value $\mathcal{R}_i(\mathbf{w}_i^{(K)}; \hat{\mathcal{D}}_i')$. The attack generator can be selected from Projected Gradient Descent (Madry et al., 2017), Fast Gradient Sign Method (Goodfellow et al., 2014), etc. Here $\epsilon$ is used to control the attack strength in the training.

---

**Algorithm S1** R-MAML$_{\text{out}}$

---

**Input:** The initialization weights $\mathbf{w}$; Distribution over tasks $p(\mathcal{T})$; Step size parameters $\alpha, \beta_1, \beta_2$.

1  **while** not done **do**
2      Sample batch of tasks $\mathcal{T}_i \sim p(\mathcal{T})$ and separate data in $\mathcal{T}_i$ into $(\mathcal{D}_i, \mathcal{D}_i')$
3      **for each** $\mathcal{T}_i$ **do**
4          $\mathbf{w}_i^{(0)} := \mathbf{w}$
5          **for** $k = 1, 2, \cdots, K$ **do**
6              $\mathbf{w}_i^{(k)} = \mathbf{w}_i^{(k-1)} - \alpha \nabla_{\mathbf{w}_i} \ell_i(\mathbf{w}_i^{(k-1)}; \mathcal{D}_i, \mathbf{w})$
7          **end for**
8          Using attack generator to generate adversarial validation data $\hat{\mathcal{D}}_i'$ by maximizing adversarial loss $\mathcal{R}_i(\mathbf{w}_i^{(K)}; \hat{\mathcal{D}}_i')$ with the constraint $\|\hat{\mathcal{D}}_i' - \mathcal{D}_i'\|_\infty \leq \epsilon$
9      **end for**
10     $\mathbf{w} := \mathbf{w} - \beta_1 \nabla_{\mathbf{w}} \sum_{\mathcal{T}_i \sim p(\mathcal{T})} \ell_i(\mathbf{w}_i^{(K)}; \mathcal{D}_i', \mathbf{w}) - \beta_2 \gamma_{\text{out}} \nabla_{\mathbf{w}} \sum_{\mathcal{T}_i \sim p(\mathcal{T})} \mathcal{R}_i(\mathbf{w}_i^{(K)}; \hat{\mathcal{D}}_i')$
11 **end while**
12 **Return:** $\mathbf{w}$

---

# 2 DETAILS OF LEARNED SIGNATURE OF NEURON'S ACTIVATION

By maximizing a single coordinate of the neuron activation vector $\mathbf{r}$ (the output before the fully-connected layer) with a perturbation in the input, the perturbation will show different behaviors between a robust model and a standard model (Engstrom et al., 2019). To be more specific, the feature pattern is revealed in the input under a robust model, while a standard model does not have such behavior. The optimization problem can be mathematically written in the following form

$$\begin{aligned} \underset{\boldsymbol{\delta}}{\text{maximize}} \quad & r_i(\mathbf{x} + \boldsymbol{\delta}) \\ \text{subject to} \quad & -\mathbf{x}_j \leq \boldsymbol{\delta}_j \leq 255 - \mathbf{x}_j, \end{aligned} \tag{S1}$$

where $r_i$ denotes the $i$-th coordinate of neuron activation vector. $\boldsymbol{\delta}$ is the perturbation in the input. $\mathbf{x}_j$ is the $j$-th pixel of the image vector $\mathbf{x}$.

# 3 VISUALIZATION OF IAMS BEFORE AND AFTER FINE-TUNING IN META-TESTING

Once obtain a model using R-MAML$_{\text{out}}$, we can test the impact of the standard fine-tuning on its robustness. Figure S1 shows a randomly selected neuron's inverted input attribution maps (IAMs) before standard fine-tuning and after standard fine-tuning in the meta-testing phase. The second row shows IAMs of the model before fine-tuning. The third row shows IAMs of the model after fine-tuning. One can find that the difference is small between the IAMs before fine-tuning and after fine-tuning, suggests that robust meta-update itself can provide the robustness adaptation without additional adversarial training.

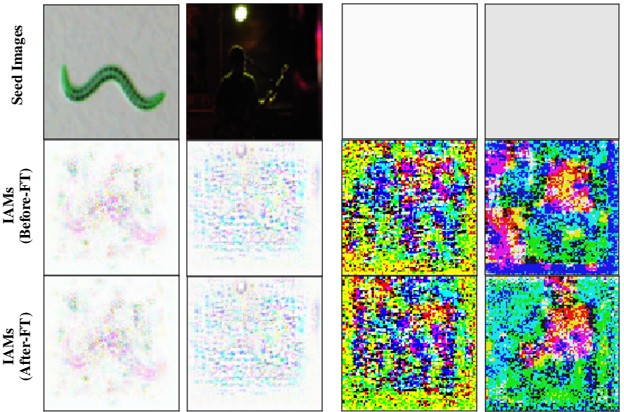

Figure S1: Visualization of a randomly selected neuron's inverted input attribution maps (IAMs) before fine-tuning and after fine-tuning in meta-testing. The model is obtained by R-MAML$_{\text{out}}$. The second row shows IAMs of the model before fine-tuning. The third row shows IAMs of the model after fine-tuning. One can find that the difference between the IAMs before fine-tuning and after fine-tuning is small, suggests that robust meta-update itself can provide the robustness adaptation without additional adversarial training.

## 4 DETAILS OF EXPERIMENTS

To test the effectiveness of our methods, we employ the MiniImageNet dataset Vinyals et al. (2016), which is the benchmark for few-shot learning. MiniImageNet contains 100 classes with 600 samples in each class. We use the training set with 64 classes and test set with 20 classes. In our experiments, we downsize each image to $84 \times 84 \times 3$.

we consider the 1-shot 5-way image classification task, i.e., the inner-gradient update (fine-tuning) is implemented using five classes and one fine-tuning image for each class in one single task. In meta-training, Each batch contains four tasks. We set the number of gradient update steps $K = 5$ in meta-training. For the meta-update, we use 15 validation images for each class. We set the gradient step size in the fine-tuning as $\alpha = 0.01$, and the gradient step sizes in the meta-update as $\beta_1 = 0.001, \beta_2 = 0.001$ for clean validation data and adversarial validation data, respectively.

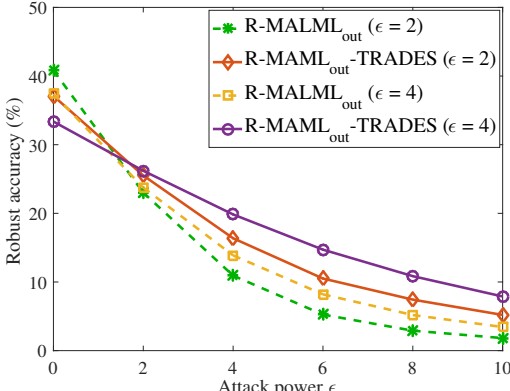

Figure S2: RA versus (testing-phase) PGD attacks at different values of perturbation strength $\epsilon$. Here the robust models are trained by R-MAML$_{\text{out}}$-TRADES and R-MAML$_{\text{out}}$. Each method trains two models under the training attack strength of $\epsilon = 2, 4$, respectively. Results show that R-MAML$_{\text{out}}$-TRADES has the ability to defend stronger attacks than R-MAML$_{\text{out}}$.

## 5 ADDITIONAL COMPARISONS ON MINIIMAGENET

Figure S2 shows robust accuracy (RA) performance of models trained using our methods. One can see that R-MAML$_{out}$-TRADES has the ability to defend stronger attacks than R-MAML$_{out}$.

In Table S1, we compare the SA/RA performance of variants of R-MAML$_{out}$ including R-MAML$_{out}$(AT), the TRADES regularization with unlabeled data R-MAML$_{out}$-TRADES, the version with contrastive learning R-MAML$_{out}$-CL. One can see that R-MAML$_{out}$-CL yields the best SA and RA among all meta-learning methods.

Table S1: SA/RA performance of different variants of proposed R-MAML$_{out}$ under the 1-shot 5-way scenario on ResNet18.

|  | SA | RA |
|---|---|---|
| MAML | 43.1% | 5.347% |
| AQ (Goldblum et al., 2019) | 30.04% | 20.05% |
| R-MAML$_{out}$(AT) (ours) | 38.94% | 19.94% |
| R-MAML$_{out}$-TRADES (ours) | 41.94% | 20.19% |
| R-MAML$_{out}$-CL (ours) | **43.98%** | **21.47%** |

## 6 EXPERIMENTS ON CIFAR-FS

We also test our proposed methods on CIFAR-FS (Bertinetto et al., 2018), which is an image classification dataset containing 64 classes of training data and 20 classes of evaluation data. The compared methods are the same as in Table 3. We keep the settings to be the same as in the test on MiniImagenet except we set $\epsilon = 8$. To perform data augmentation in experiments, we mine 500 additional unlabeled data for each training class from the STL-10 dataset (Coates et al., 2011).

Table S2 and Table S3 show the comparisons in 1-Shot 5-Way and 5-Shot 5-Way learning scenarios, respectively. One can see that our methods outperform the baseline methods MAML and AQ (Goldblum et al., 2019). The results also indicate that semi-supervised learning (in terms of TRADES and contrastive learning) can further boost the performance. In particular, as shown by Table S2 and Table S3, R-MAML$_{out}$-CL leads to 10% SA improvement and 3% RA improvement compared to AQ under the MAML 1-Shot 5-Way setting, and 5.6% SA improvement and 1.3% RA improvement under the 5-Shot 5-Way setting.

Table S2: SA/RA performance of our proposed methods on CIFAR-FS (Bertinetto et al., 2018) (1-Shot 5-Way).

|  | SA | RA |
|---|---|---|
| MAML | **51.07%** | 0.235% |
| AQ (Goldblum et al., 2019) | 31.25% | 26.34% |
| R-MAML$_{out}$(AT) (ours) | 39.76% | 26.15% |
| R-MAML$_{out}$(TRADES) (ours) | 40.23% | 27.45% |
| R-MAML$_{out}$-TRADES (ours) | 40.59% | 28.06% |
| R-MAML$_{out}$-CL (ours) | 41.25% | **29.33%** |

## 7 EXPERIMENTS ON OMNIGLOT

We then conduct experiments on Omniglot (Lake et al., 2015), which includes handwritten characters from 50 different alphabets. There are 1028 classes of training data and 423 classes of evaluation

Table S3: SA/RA performance of our proposed methods on CIFAR-FS (Bertinetto et al., 2018) (5-Shot 5-Way).

| | SA | RA |
|---|---|---|
| MAML | **67.2%** | 0.225% |
| AQ (Goldblum et al., 2019) | 52.32% | 33.96% |
| R-MAML$_{out}$(AT) (ours) | 57.18% | 32.62% |
| R-MAML$_{out}$(TRADES) (ours) | 57.46% | 34.72% |
| R-MAML$_{out}$-TRADES (ours) | 57.62% | 34.76% |
| R-MAML$_{out}$-CL (ours) | 57.95% | **35.30%** |

data. Due to the hardness of finding the unlabeled data with similar patterns, we only test our supervised learning methods on Omniglot. We compare R-MAML$_{out}$(TRADES) to AQ (Goldblum et al., 2019) in the 1-shot (5, 10, 15, 20)-Way settings. Figure. S3 shows the results of RA/SA under $\epsilon = 10$. The results show that R-MAML$_{out}$(TRADES) can obtain better performance than AQ.

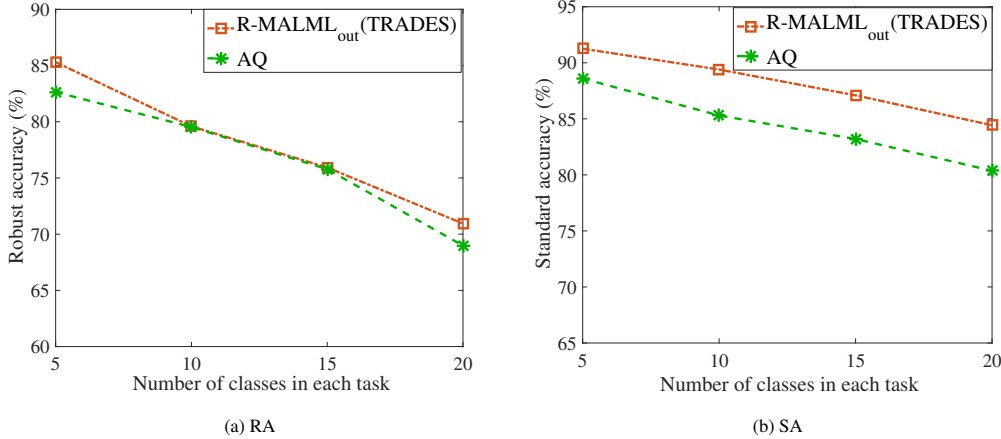

(a) RA         (b) SA

Figure S3: Performance of R-MAML$_{out}$(TRADES) and AQ (Goldblum et al., 2019) on Omniglot versus number of classes in each task (from 5 to 20 ways): (a) RA. (b) SA.

