# OpenReview forum: "On Fast Adversarial Robustness Adaptation in Model-Agnostic Meta-Learning"
_ICLR.cc/2021/Conference — ICLR 2021 Poster_

### Official Review · AnonReviewer3 · 2020-10-28
**submission 2301 review**

**Rating:** 6
**Confidence:** 2

**Review:**

The paper proposes to integrate meta-learning (MAML specifically) and adversarial training (AT) to improve adversarial robustness of meta-learning in terms of fast and effective adaptation on few-shot test tasks. To achieve this, the authors provide extensive investigation and solid answers on when, how and why their method works.

Overall I vote for accepting. The authors found that it seems to be natural to combine MAML and AT together to enhance the adversarial robustness of meta-learning, however, due to computational complexity and bi-level optimization in MAML and AT, they further pose the above three questions when, how, and why step by step, to which they also provide affirmative answers in the paper. Some suggestions are given in the cons part below.

Pros:

1. Overall the paper is well-written. In particular, its motivation and contribution are clearly explained and summarized, which is quite fluent when reading.

2. The paper presents visual evidence of when to incorporate robustness regularization in MAML by leveraging input attribution maps of neurons, which is reasonable and interesting for me.

3. The paper leverages contrastive learning into MAML to help the model’s adversarial robustness, which is novel and interesting to me.

Cons:

1. For the experiment part, more other few-shot learning datasets (such as Omniglot) or other few-shot settings (such as 5-way or 20-way) can be utilized to evaluate the proposed method as well as to present more data points.

---

> ### Author Response · Authors · 2020-11-19
> **Our Response to Reviewer 3**
>
> We thank the reviewer very much for the positive rating on our work and  the insightful suggestions. We also add the suggested experiments to further address your question; see details below. Please don’t hesitate to let us know for any additional comments on the paper or on the changes.
>
> **Q:** *More other few-shot learning datasets (such as Omniglot) or other few-shot settings (such as 5-way or 20-way).*
>
> **A:** Thank you for the suggestion. We conducted additional experiments on Omniglot [1] (Figure S3 in the revised paper over ${5,10,15,20}$-way setups) as well as  CIFAR-FS [2] (Tables S2, S3).
>
> 1. As shown in Table S2 and S3 (copied below), our methods outperform the baseline methods MAML and AQ in robust accuracy (RA) and/or standard accuracy (SA). For example,  the proposed R-MAML$_\mathrm{out}$-CL leads to 10% improvement in SA and 3% improvement in RA compared to AQ under the MAML 1-Shot 5-Way setting, and 5.6% SA improvement and 1.3% RA improvement under the 5-Shot 5-Way setting.
>
> 2. We also conducted new experiments in more few-shot settings. Experiments on 1-Shot 5-Way and 5-Shot 5-Way are conducted on CIFAR-FS (Tables S2, S3). On Omniglot, we show R-MAML$_\mathrm{out}$(TRADES) can obtain better performance than AQ in the 1-Shot (5,10,15,20)-Way settings (see Figure S3 in the revised paper).
>
> **Table S2.** SA/RA performance of our proposed methods on CIFAR-FS (1-Shot 5-Way, $\epsilon = 8$).
>
> || Standard Accuracy (SA) &nbsp; | Robust Accuracy (RA) |
> |:-:|:-:|:-:|
> |MAML|**51.07%**|0.235%|
> |AQ|31.25%|26.34%|
> |R-MAML$_\mathrm{out}$(AT) (ours)|39.76%|26.15%|
> |R-MAML$_\mathrm{out}$(TRADES) (ours)|39.76%|26.15%|
> |R-MAML$_\mathrm{out}$-TRADES (ours)|40.59%|28.06%|
> |R-MAML$_\mathrm{out}$-CL (ours)|41.25%|**29.33%**|
>
> **Table S3.** SA/RA performance of our proposed methods on CIFAR-FS (5-Shot 5-Way, $\epsilon = 8$).
>
> || Standard Accuracy (SA) &nbsp; | Robust Accuracy (RA) |
> |:-:|:-:|:-:|
> |MAML|**67.2%**|0.225%|
> |AQ|52.32%|33.96%|
> |R-MAML$_\mathrm{out}$(AT) (ours)|57.18%|32.62%|
> |R-MAML$_\mathrm{out}$(TRADES) (ours)|57.46%|34.72%|
> |R-MAML$_\mathrm{out}$-TRADES (ours)|57.62%|34.76%|
> |R-MAML$_\mathrm{out}$-CL (ours)|57.95%|**35.3%**|
>
> [1] Lake et al., “Human-level concept learning through probabilistic program induction”, 2015
>
> [2] Bertinetto et al., “Meta-learning with differentiable closed-form solvers”, 2018

---

### Official Review · AnonReviewer2 · 2020-10-29
**Clear and interesting paper but may not be enough convincing**

**Rating:** 6
**Confidence:** 4

**Review:**

It is an interesting paper empirically addressing adversarial robustness of model agnostic meta learning (MAML). The paper investigates where to incorporate robust regularization in MAML in order to improve adversarial robustness, and based on that *efficient* robust MAML methods are proposed. Interestingly, contrastive learning is incorporated and derive a more robust MAML model.

My concerns are listed as below:

- The paper is highly empirical, whearas only one dataset is employed. The claims would be more convincing with more datasets.
- The key point is not properly emphasized. For example, Section 4 and Section 5.1 (TRADES) do not provide very important insights compared with Section 5.2 (CL) — maybe Section 5.2  could be extended furthur with some other contents defered in the Appendix.

---

> ### Author Response · Authors · 2020-11-19
> **Our Response to Reviewer 2**
>
> We thank the review for the encouraging feedback. We list our response below. Please don’t hesitate to let us know for any additional comments.
>
> **Q1:** *The claims would be more convincing with more datasets.*
>
> **A1:** Thank you for the suggestion. We conducted additional experiments on CIFAR-FS [1] (Table S2 and S3 in the revised paper) and Omniglot [2]  (Figure S3 in the revised paper under various few-shot settings). The results show that our methods work well on various datasets and outperform the baseline methods. For example, as shown in Table S2 and S3 (copied below), the use of unlabeled data augmentation (R-MAML$_\mathrm{out}$-CL) on CIFAR-FS can provide 10% (or 5.6%) standard accuracy (SA) improvement and 3% (or 1.3%) robust accuracy (RA) improvement over AQ under the 1-Shot 5-Way (or 5-Shot 5-Way) setting.
>
> **Table S2.** SA/RA performance of our proposed methods on CIFAR-FS (1-Shot 5-Way, $\epsilon = 8$).
>
> || Standard Accuracy (SA) &nbsp; | Robust Accuracy (RA) |
> |:-:|:-:|:-:|
> |MAML|**51.07%**|0.235%|
> |AQ|31.25%|26.34%|
> |R-MAML$_\mathrm{out}$(AT) (ours)|39.76%|26.15%|
> |R-MAML$_\mathrm{out}$(TRADES) (ours)|39.76%|26.15%|
> |R-MAML$_\mathrm{out}$-TRADES (ours)|40.59%|28.06%|
> |R-MAML$_\mathrm{out}$-CL (ours)|41.25%|**29.33%**|
>
> **Table S3.** SA/RA performance of our proposed methods on CIFAR-FS (5-Shot 5-Way, $\epsilon = 8$).
>
> || Standard Accuracy (SA) &nbsp; | Robust Accuracy (RA) |
> |:-:|:-:|:-:|
> |MAML|**67.2%**|0.225%|
> |AQ|52.32%|33.96%|
> |R-MAML$_\mathrm{out}$(AT) (ours)|57.18%|32.62%|
> |R-MAML$_\mathrm{out}$(TRADES) (ours)|57.46%|34.72%|
> |R-MAML$_\mathrm{out}$-TRADES (ours)|57.62%|34.76%|
> |R-MAML$_\mathrm{out}$-CL (ours)|57.95%|**35.3%**|
>
> [1] Bertinetto et al., “Meta-learning with differentiable closed-form solvers”, 2018
>
> [2] Lake et al., “Human-level concept learning through probabilistic program induction”, 2015
>
> **Q2:** *The key point is not properly emphasized. For example, Section 4 and Section 5.1 (TRADES) do not provide very important insights compared with Section 5.2 (CL) — maybe Section 5.2 could be extended further with some other contents deferred in the Appendix.*
>
> **A2:** Thanks for the suggestion. We would like to re-emphasize the important insights from Sec. 4 and Sec. 5.1 (TRADES).
>
> 1. The studies in Sec. 4 were motivated to address the problem “how to efficiently train robustness-regularized MAML?”. This is a critical problem since both MAML and adversarial training take extensive computational complexities, where MAML is in the form of bi-level optimization and adversarial training is given by a min-max optimization problem. We addressed this problem by leveraging 1) fast adversarial example generation (namely, FGSM), and 2) partial model fine-tuning (namely, ANIL). We showed in Sec. 4 that both FGSM and ANIL can improve the computation efficiency of robust MAML, which takes the similar training cost to the standard MAML (Table 2). Moreover, compared to ANIL, FGSM-enabled robust MAML yields the most graceful tradeoff between computation efficiency, robustness and accuracy.
>
> 2. The studies in Sec. 5.1 (TRADES) is a ‘must-try’ step when generalizing robustness-regularized MAML from supervision to semi-supervision, since TRADES is directly applicable to data augmentation and has been used in state-of-the-art adversarial defenses [3,4]. Indeed, we showed that for TRADES-enabled MAML that uses unlabeled data augmentation can improve the robustness adaptation in MAML (Figure 3). Furthermore, TRADES-enabled MAML provides us a semi-supervised baseline when comparing it with CL-enabled MAML. To the best of our knowledge, the semi-supervised robustness-promoting MAML has not yet been studied.
>
> 3. In addition to Sec. 4 and Sec. 5, Sec. 2 and 3 also offer us important insights.
> - In Sec. 2, we propose a principled problem formulation  (3), which covers the baseline approach adversarial querying (AQ) (Goldblum et al., 2019) as a special case. As we clarified in the last paragraph of page 5, our proposed problem is simplified to AQ as  $\gamma_{\mathrm{in}} = 0$ and  $\gamma_{\mathrm{out}} = \infty $. The formulation (3) also motivates us to ‘when’ to incorporate robustness regularization in MAML (inner level vs. outer level).
> - In Sec. 3,  we show that it is sufficient to promote adversarial robustness at the meta-update level. Most importantly, we explain our conclusion from two insightful perspectives, 1) explanation of learned neuron’s representation; and 2) resilience to different fine-tuning schemes at the meta-testing phase.
>
> [3] Uesato, Jonathan, et al. "Are labels required for improving adversarial robustness?." arXiv preprint arXiv:1905.13725 (2019).
>
> [4] Zhang, Hongyang, et al. "Theoretically principled trade-off between robustness and accuracy." arXiv preprint arXiv:1901.08573 (2019).

---

### Official Review · AnonReviewer4 · 2020-10-30
**clear motivation and reasonable method, but marginal novelty**

**Rating:** 6
**Confidence:** 4

**Review:**

This paper explores a way to promote the adversarial robustness in Model-agnostic meta-learning (MAML). It conducts extensive experiments to show regularizing adversarial robustness at meta-update level is sufficient to offer fast and effective robustness adaptation on few-shot test tasks. However, it lacks the theoretical analysis for this conclusion. Also, the experiments only are conducted on few-shot image classification task on miniImageNet dataset. This makes this conclusion lack sufficient credibility.

This paper also shows the use of unlabeled data augmentation can provide additional benefits on adversarial robustness of MAML. Due to the low-data regime in few-shot classification, it is reasonable to introduce contrastive learning task to few-shot learning, which may facilitate the development of following research.

It is of great importance to investigate the adversarial robustness in meta-learning framework, such as MAML in this paper. It might give us insights on designing network structure. However, it would be better to give more insights on the way to integrate the adversarial robustness rather than list the experimental results.

Novelty of this paper seems to be technically marginal. While the aggregation of adversarial robustness on meta-learning framework is new, detailed architecture explanation with novelty is lacked.

---

> ### Author Response · Authors · 2020-11-19
> **Our Response to Reviewer 4 - Part 2**
>
> **Q3:** *It would be better to give more insights on the way to integrate the adversarial robustness rather than list the experimental results.*
>
> **A3:** Thank you for the suggestion. We summarize our findings and list the related insights.
>
> 1. In Section 2 - Robustness-promoting MAML, we propose a principled robustness-regularized meta-learning framework in (3), which covers the baseline approach adversarial querying (AQ) (Goldblum et al., 2019) as a special case. As we clarified in the last paragraph of page 5, our proposed problem is simplified to AQ as  $\gamma_{\mathrm{in}} = 0$ and  $\gamma_{\mathrm{out}} = \infty$.
>
> 2. In Section 3, motivated by (3), we address the question of when to incorporate robust training in MAML (outer meta-learning level vs. inner fine-tuning level) .
> Our empirical finding is that it is sufficient to promote adversarial robustness at the meta-update level.  Most importantly, we explain our conclusion from two insightful perspectives, 1) explanation of learned neuron’s representation; and 2) resilience to different fine-tuning schemes at the meta-testing phase.
>
> 3. In Section 4, we then address the question “how to efficiently train robustness-regularized MAML?” Note that robust training is given by a min-max optimization, and MAML is given by a bi-level optimization. Thus, the computation efficiency is a critical question. We addressed this problem by leveraging 1) fast (one-step) adversarial example generation, and 2) partial model training during fine-tuning (only fine-tuning the classifier’s head).
>
> 4. In Section 5, we find that proper unlabeled data augmentation can further improve the accuracy-robustness tradeoff in MAML  (Figure 3). In particular, we clearly motivate why self-supervised contrastive representation learning is favored in MAML (see paragraph after Figure 3). And we show that self-supervised contrastive representation learning is beneficial to MAML in both robustness and accuracy (Table 3).
>
> **Q4:** *Detailed architecture explanation with novelty is lacking.*
>
> **A4:** We would like to remark that our proposed methods do not rely on the specific model architecture. In the paper, we show that our methods obtain good performance under both four-layer CNN (see Table 3) and ResNet18 (see Table S1 in the supplementary). As we responded to the reviewer's first question, we believe that the lack of theoretical analysis does not restrict our contributions. Our technical contribution is also not marginal. The proposed optimization framework (3) and the solutions to scalable training of robustness-regularized MAML and semi-supervision enabled robustness-regularized MAML are all technically new to the meta-learning.
>
> We hope that our responses have mostly addressed your concerns. If you have further comments, please feel free to let us know. We will try our best to address them.

---

> ### Author Response · Authors · 2020-11-19
> **Our Response to Reviewer 4 - Part 1**
>
> We thank the reviewer for the suggestions and comments. We reply individually to each raised point below. Please feel free to let us know if you have additional comments.
>
> **Q1:** *Marginal novelty and lacks theoretical analysis.*
>
> **A1:** Even if our conclusion is drawn based on empirical results, we do not think that this limits our novelty. In fact, we have tried our best to offer motivations and insights behind our approach and empirical results. For example,
> - In Sec. 3, we have justified our conclusion on the sufficiency of adversarial robustness at meta-update level from two insightful perspectives, 1) explanation of learned neuron’s representation; and 2) resilience to different fine-tuning schemes at the meta-testing phase.
> - In Sec. 4, we contributed to scalable training. Computation efficiency is critical since both robust training (a min-max optimization problem) and MAML (a bi-level optimization problem) are involved in robust MAML. We found 1) fast (one-step) adversarial example generation and 2) partial model training (only fine-tuning the classifier’s head) effective in training acceleration (30% speed-up) without losing much performance.
> - In Sec. 5, we have generalized the robust MAML to the semi-supervised setting. We have shown that the use of unlabeled data augmentation, particularly introducing an auxiliary contrastive learning task, can provide additional benefits on adversarial robustness of MAML in the low data regime: 2% robust accuracy improvement and 9% standard accuracy improvement over the baseline method (Table 3). Additional justifications on CIFAR-FS have also been provided in Table S2 and S3; see the revised paper or Q2-A2.
>
> On the other hand, spurred by the reviewer’s suggestion, we review more literature to track the state-of-the-art theoretical analysis in solving problems of MAML and adversarial training (AT). Unfortunately, both directions have very limited progress in theory, since both MAML and AT are in the form of quite challenging optimization problems, where the former is given by a bi-level optimization problem and the latter is given by a min-max optimization problem.  The existing theoretical analysis for MAML and AT [1,2,3] is restricted to the convergence rate of bi-level and min-max optimization algorithms under strong assumptions, e.g. convexity required for the inner-level optimization problem.  What is more, our setting becomes more difficult to analyze as AT (min-max optimization) is embedded in MAML (bi-level optimization). Thus, although having theoretical analysis is attractive, it is much beyond the scope of our work. We will add this discussion in a further work session.
>
> [1] Hong et al., “A Two-Timescale Framework for Bilevel Optimization: Complexity Analysis and Application to Actor-Critic”, 2020
>
> [2] Fallah et al., "On the convergence theory of gradient-based model-agnostic meta-learning algorithms." International Conference on Artificial Intelligence and Statistics. 2020.
>
> [3] Gao, Ruiqi, et al. "Convergence of adversarial training in overparametrized neural networks." Advances in Neural Information Processing Systems. 2019.
>
> **Q2:** *The experiments only are conducted on few-shot image classification tasks on miniImageNet dataset. This makes this conclusion lack sufficient credibility.*
>
> **A2:** Thank you for the suggestion. We conducted additional experiments on CIFAR-FS [4] (Table S2 and S3) and Omniglot [5]  (Figure S3 in our revised paper across different few-shot setups). The results show that our methods work well on various datasets and outperform the baseline methods. For example, as shown in Table S2 and S3, the use of unlabeled data augmentation (R-MAML$_\mathrm{out}$-CL) on CIFAR-FS can provide 10% (or 5.6%) standard accuracy (SA) improvement and 3% (or 1.3%) robust accuracy (RA) improvement over AQ under the 1-Shot 5-Way (or 5-Shot 5-Way) setting.
>
> **Table S2.** SA/RA performance of our proposed methods on CIFAR-FS (1-Shot 5-Way, $\epsilon = 8$).
>
> || Standard Accuracy (SA) &nbsp; | Robust Accuracy (RA) |
> |:-:|:-:|:-:|
> |MAML|**51.07%**|0.235%|
> |AQ|31.25%|26.34%|
> |R-MAML$_\mathrm{out}$(AT) (ours)|39.76%|26.15%|
> |R-MAML$_\mathrm{out}$(TRADES) (ours)|39.76%|26.15%|
> |R-MAML$_\mathrm{out}$-TRADES (ours)|40.59%|28.06%|
> |R-MAML$_\mathrm{out}$-CL (ours)|41.25%|**29.33%**|
>
> **Table S3.** SA/RA performance of our proposed methods on CIFAR-FS (5-Shot 5-Way, $\epsilon = 8$).
>
> || Standard Accuracy (SA) &nbsp; | Robust Accuracy (RA) |
> |:-:|:-:|:-:|
> |MAML|**67.2%**|0.225%|
> |AQ|52.32%|33.96%|
> |R-MAML$_\mathrm{out}$(AT) (ours)|57.18%|32.62%|
> |R-MAML$_\mathrm{out}$(TRADES) (ours)|57.46%|34.72%|
> |R-MAML$_\mathrm{out}$-TRADES (ours)|57.62%|34.76%|
> |R-MAML$_\mathrm{out}$-CL (ours)|57.95%|**35.3%**|
>
> [4] Bertinetto et al., “Meta-learning with differentiable closed-form solvers”, 2018
> [5] Lake et al., “Human-level concept learning through probabilistic program induction”, 2015

---

### Official Review · AnonReviewer1 · 2020-11-01
**explores adversarial robustness and contrastive learning of MAML**

**Rating:** 6
**Confidence:** 4

**Review:**

Summary

This paper investigates the adversarial robustness of model agnostic meta-learning (MAML). Adversarial robustness can be added to MAML in two places, meta-update stage and and fine-tune stage. It shows that robustifying the meta-update stage via fast attack generation method is sufficient to achieve fast robustness adaptation without losing generalization and computation efficiency in general. The paper also demonstrates that unlabeled data can help using contrastive representation learning to improve generalization and robustness.

Strengths

It shows:
1. adversarial training with the projected gradient descent (PGD) attack generation method is only needed at meta-update stage for MAML.

2. computationally efficient adversarial training methods such as  almost-no-inner-loop (ANIL) fine-tuning strategy and FGSM are enough.

3. training with unlabeled data using contrastive learning further improves generalization and robustness.

Weaknesses

Adversarial robustness of MAML has been studied in (Goldblum et al., 2019). This paper provides further investigation. The study seems to be straightforward and incremental.

All results are based on miniImageNet. MAML has also been applied in reinforcement learning. It is not clear the results in this paper carries out to the RL setting.

Decision

The paper improves MAML with adversarial training at meta-update stage and unlabeled data through contrastive learning. However, the studies and techniques seem to be incremental compared with related work.


=====POST-REBUTTAL COMMENTS========

The authors provided additional experiments on CIFAR-FS and Omniglot. The results show that their methods outperform the baseline method adversarial querying (AQ).

It is still not clear whether the methods work in the reinforcement learning setting. As the original MAML paper shows that MAML works for RL problems, it is important to address this question. Otherwise, it could potentially limits the applicability of the proposed methods in the paper.

I still have concerns over their novelty and the significance of their contributions.

Overall, I applaud their effort to address my comments. I am more positive on the paper than before. My rating is a solid 6. The paper in the current stage does not warrant a higher rating for ICLR in my opinion.

---

> ### Author Response · Authors · 2020-11-19
> **Our Response to Reviewer 1 - Part 2**
>
> **Q2:** *All results are based on MiniImageNet. MAML has also been applied in Reinforcement Learning (RL). It is not clear the results in this paper carry out to the RL setting.*
>
> **A2:** Thank you for the suggestion. We conducted additional experiments on CIFAR-FS [1] (see Tables S2 and S3) and Omniglot [2]  (see Figure S3 in Appendix 7 of the revised paper revision). The results show that our methods consistently work well on various datasets and setups and outperform the baseline method AQ. In particular, as shown by Table S2 and Table S3, the proposed R-MAML$_\mathrm{out}$-CL leads to 10% SA improvement and 3% RA improvement compared to AQ under the MAML 1-Shot 5-Way setting, and 5.6% SA improvement and 1.3% RA improvement under the 5-Shot 5-Way setting.
>
> For the comment about applying MAML to Reinforcement learning, we believe that our proposed problem formulation in (3), together with its training method and insights, could help solve problems in the generic form of min-max involved bi-level optimization. However, since in this paper we mainly focused on the tasks of image classification, we will leave it as a future research direction and add a future work section in the revision. Thank you for your great comment! We will also release our code so that future studies on RL+MAML can benefit from our works.
>
> **Table S2.** SA/RA performance of our proposed methods on CIFAR-FS (1-Shot 5-Way, $\epsilon = 8$).
>
> || Standard Accuracy (SA) &nbsp; | Robust Accuracy (RA) |
> |:-:|:-:|:-:|
> |MAML|**51.07%**|0.235%|
> |AQ|31.25%|26.34%|
> |R-MAML$_\mathrm{out}$(AT) (ours)|39.76%|26.15%|
> |R-MAML$_\mathrm{out}$(TRADES) (ours)|39.76%|26.15%|
> |R-MAML$_\mathrm{out}$-TRADES (ours)|40.59%|28.06%|
> |R-MAML$_\mathrm{out}$-CL (ours)|41.25%|**29.33%**|
>
> **Table S3.** SA/RA performance of our proposed methods on CIFAR-FS (5-Shot 5-Way, $\epsilon = 8$).
>
> || Standard Accuracy (SA) &nbsp; | Robust Accuracy (RA) |
> |:-:|:-:|:-:|
> |MAML|**67.2%**|0.225%|
> |AQ|52.32%|33.96%|
> |R-MAML$_\mathrm{out}$(AT) (ours)|57.18%|32.62%|
> |R-MAML$_\mathrm{out}$(TRADES) (ours)|57.46%|34.72%|
> |R-MAML$_\mathrm{out}$-TRADES (ours)|57.62%|34.76%|
> |R-MAML$_\mathrm{out}$-CL (ours)|57.95%|**35.3%**|
>
> [2] Lake et al., “Human-level concept learning through probabilistic program induction”, 2015

---

> ### Author Response · Authors · 2020-11-19
> **Our Response to Reviewer 1 - Part 1**
>
> We thank the reviewer very much for the insightful comments. We list pointwise responses to your questions below. Please feel free to let us know if you have additional comments.
>
> **Q1:** *Adversarial robustness of MAML has been studied in (Goldblum et al., 2019). This paper provides further investigation. The study seems to be straightforward and incremental.*
>
> **A1:** The differences between our work and the method in (Goldblum et al., 2019) are in **threefolds**.
>
> 1. We have  provided insights and explanation to why the adversarial meta-update is critical from two perspectives while (Goldblum et al., 2019) does not (as also recognized by Reviewer 3). Our insights are based on:
> (a) learned data feature representation (paragraph “Learned signature of neuron’s representation” at page 5)
> (b) robustness adaptation in meta-testing (paragraph “robust meta-update provides robustness adaptation” at page 5).
> Besides,  as we clarified in the last paragraph of page 5, adversarial querying (AQ) (Goldblum et al., 2019) is actually a *special case* of our proposed general framework (3) with  $\gamma_{\mathrm{in}} = 0$ and  $\gamma_{\mathrm{out}} = \infty $. Unlike our proposal,  the meta-update of AQ is overridden by the AT regularization without incorporating the standard validation loss. As a result, AQ leads to a worse clean accuracy than our proposed R-MAML$_\mathrm{out}$; see Tables 3 in our paper and Tables 6, 21 in (Goldblum et al., 2019). The newly added experiment also supports this conclusion (see Tables S2 - S3 in Q2-A2, and Figure S3 in the revision).
>
> 2. As training a robustness-regularized MAML is very expensive, in this paper we propose a more efficient training for robustness-regularized MAML while (Goldblum et al., 2019) does not. Computation efficiency is a critical problem in robustness-regularized MAML, since we have both robust training (a min-max optimization problem) and MAML (a bi-level optimization problem) at the same time. We addressed this problem by leveraging 1) fast (one-step) adversarial example generation, and 2) partial model training during fine-tuning (only fine-tuning the classifier’s head). The speed can be increased by 30% using these acceleration techniques. Please see Sec. 4 for more details.
>
> 3. We have generalized the robust MAML to the semi-supervised setting while (Goldblum et al., 2019) does not. In Sec. 5, we show that the use of unlabeled data augmentation, particularly introducing an auxiliary contrastive learning task, can provide additional benefits on adversarial robustness of MAML in the low data regime: 2% robust accuracy improvement and 9% clean accuracy improvement over AQ (Table 3). Furthermore, in the revision, we have conducted additional experiments and show that the use of unlabeled data augmentation on CIFAR-FS [1] can provide 10% (or 5.6%) standard accuracy (SA) improvement and 3% (or 1.3%) robust accuracy (RA) improvement over AQ under the 1-Shot 5-Way (or 5-Shot 5-Way) setting (see Table S2 and S3 in the next response - Part 2).
>
> [1] Bertinetto et al., “Meta-learning with differentiable closed-form solvers”, 2018

---

### Author Response · Authors · 2020-11-19
**Response Highlights**

We thank all reviewers for their time and efforts in reviewing the paper and providing helpful suggestions/comments. The major changes have been highlighted in blue in the revision. In what follows, we highlighted our contributions and newly added experiments suggested by reviewers. Details will be provided in point-wise responses.

**Contribution**

In our paper, we propose a general framework for robustness-regularized meta-learning (Section 2). We also provide insights and step-by-step investigations to show when to incorporate robust training and why it works (Section 3). Noticing the long running time of the MAML inner-loop update and adversarial example generation for the outer-loop, we have made additional contributions on scalable training (Section 4). Finally, we show that incorporating unlabeled data and contrastive learning can further enhance the model’s robustness (Section 5).

**Added experiments**
1. In the revised paper, we added new experiments on CIFAR-FS [1] (See Tables S2, S3) and Omniglot [2] (See Figure S3). The results justify the improvement of our approaches over the baseline method across various datasets.
2. We added new experiments on various few-shot settings, e.g. 1-Shot 5-Way and 5-Shot 5-Way on CIFAR-FS (See Tables S2, S3), and 1-Shot (5, 10,15,20)-Way on Omniglot (See Figure S3). The results show that our proposed methods keep effective in different few-shot learning regimes.

[1] Bertinetto et al., “Meta-learning with differentiable closed-form solvers”, 2018

[2] Lake et al., “Human-level concept learning through probabilistic program induction”, 2015

---

### Decision · Program_Chairs · 2021-01-07
**Final Decision**

**Decision:**

Accept (Poster)

**Comment:**

The reviewers’ main concern was a lack of experiments, and additional experiments were provided by the authors.  While the rebuttal was not addressed by the reviewers, the AC feels that the rebuttal did address a number of experimental concerns well enough to justify accepting this paper.